# Critical Drug Loss Induced by Silicone and Polyurethane Implantable Catheters in a Simulated Infusion Setup with Three Model Drugs

**DOI:** 10.3390/pharmaceutics13101709

**Published:** 2021-10-16

**Authors:** Nicolas Tokhadzé, Philip Chennell, Bruno Pereira, Bénédicte Mailhot-Jensen, Valérie Sautou

**Affiliations:** 1Université Clermont Auvergne, CHU Clermont Ferrand, Clermont Auvergne INP, CNRS, ICCF, F-63000 Clermont-Ferrand, France; nicolas.tokhadze@gmail.com (N.T.); vsautou@chu-clermontferrand.fr (V.S.); 2CHU Clermont-Ferrand, Unité de biostatistiques, DRCI, F-63000 Clermont-Ferrand, France; bpereira@chu-clermontferrand.fr; 3Université Clermont Auvergne, ICCF, F-63000 Clermont-Ferrand, France; benedicte.mailhot@uca.fr

**Keywords:** infusion line, drug loss, drug analysis, sorption, catheters, surface characterization

## Abstract

Silicone and polyurethane are biocompatible materials used for the manufacture of implantable catheters, but are known to induce drug loss by sorption, causing potentially important clinical consequences. Despite this, their impact on the drugs infused through them is rarely studied, or they are studied individually and not part of a complete infusion setup. The aim of this work was to experimentally investigate the drug loss that these devices can cause, on their own and within a complete infusion setup. Paracetamol, diazepam, and insulin were chosen as models to assess drug sorption. Four commonly used silicone and polyurethane catheters were studied independently and as part of two different setups composed of a syringe, an extension set, and silicone or polyurethane implantable catheter. Simulated infusion through the catheter alone or through the complete setup were tested, at flowrates of 1 mL/h and 10 mL/h. Drug concentrations were monitored by liquid chromatography, and the silicone and polyurethane materials were characterized by ATR-IR spectroscopy and Zeta surface potential measurements. The losses observed with the complete setups followed the same trend as the losses induced individually by the most sorptive device of the setup. With the complete setups, no loss of paracetamol was observed, but diazepam and insulin maximum losses were respectively of 96.4 ± 0.9% and 54.0 ± 5.6%, when using a polyurethane catheter. Overall, catheters were shown to be the cause of some extremely high drug losses that could not be countered by optimizing the extension set in the setup.

## 1. Introduction

The parenteral administration of drugs into the vascular system through the skin requires the use of an infusion line, which is composed of many different medical devices (MDs), from the syringe or medication bag to the catheter, including potentially infusion and extension sets, on-line filters, etc. [1]. The infused drug will therefore come into contact successively with numerous different polymers making up the devices. It has now been largely demonstrated that these polymers can interact with the drugs, leading to drug loss by sorption or release of unwanted and potentially toxic compounds (e.g., plasticizers released from polyvinyl chloride (PVC) tubings) [2].

Overall, drug loss by sorption can result from two main mechanisms [3]: adsorption, which corresponds to the interactions of the compounds with the surface of the material, and absorption, which is the migration of the previously adsorbed compounds into the polymer matrix. As these phenomena can be responsible for potentially critical drug under-dosing, or even for a complete loss of activity in the worst case, it is therefore important to minimize the risk of drug sorption during infusion. Up until now, most of the published studies investigated the sorption phenomena between drugs and infusion or extension lines [4,5,6,7,8,9,10,11], which do not come into direct contact with the patient and generally represent, due to their length (tubings ranging up to more than 200 cm long), the most important surface area of polymer in contact with the infused drugs. A wide choice of tubings of different materials (plasticized PVC, polyurethane, polyethylene, or coextruded materials) is now commercialized, and some of them have proved to be able to markedly decrease the risk of drug loss by sorption. Indeed, in a previously published work [12], we highlighted that coextruded PE/PVC extension sets induced overall less sorption phenomena compared to PVC extension sets. However, in clinical situations, these medical devices are not used alone, but as part of an infusion setup composed of medical devices connected to each other and finishing with the catheter. Catheters are of tantamount importance as they are implanted through the skin into the blood circulation and actually administer the drugs into the patients’ body. Unfortunately, they are almost exclusively made of polyurethane (PUR) or silicone, and these materials, despite their great biocompatibility, have been shown to be at high risk of causing drug loss by sorption [13,14,15,16,17,18], but these studies are few, not recent, and do not take into account the whole infusion setup.

Therefore, this study aimed to assess the drug loss by sorption that could be induced individually by four commonly used catheters, and evaluate the impact of a complete infusion line, composed of a syringe, two different extension sets, and a catheter on the amount of drug that would be withheld by the MDs during simulated infusions.

## 2. Materials and Methods

### 2.1. Materials

#### 2.1.1. Medical Devices

All the medical devices used in this study and their characteristics are presented in Table 1. Four catheters were studied: Turbo-Flo^®^ and PowerPicc^®^ were peripherally inserted central catheters (PICC), Blue FlexTip^®^ was a central venous catheter (CVC), and Lifecath^®^ was a tunneled CVC. The syringes and catheters were purchased from their respective providers and extension sets were graciously provided by CAIR LGL.

#### 2.1.2. Medications

The following medications were used:Valium^®^ (Diazepam) 10 mg/2mL (Roche, Rosny-Sous-Bois, France; batch F1126F01, expiring 9/2020), diluted to 0.2 mg/mL in a 5% glucose solution (B. Braun, Saint Cloud, Germany).NOVORAPID^®^ (Insulin aspart) 100 UI/mL (Novo Nordisk, Courbevoie, France; batch HS65E14, expiring 1/2020), diluted to 0.1 UI/mL in a 0.9% sodium chloride solution (Versylene^®^, Fresenius Kabi, Louviers, France). Insulin aspart will henceforth be referred to as insulin.Paracetamol B BRAUN^®^ (paracetamol) 10 mg/mL (B. Braun, Saint Cloud, France; batch 18105452, expiring 2/2020 and 18141450, expiring 3/2020), diluted to 1 mg/mL in a 0.9% sodium chloride solution (Versylene^®^, Fresenius Kabi, Louviers, France).

#### 2.1.3. Reagents

The following reagents were used for chromatographic separation: acetonitrile (ACN) 99% purity (Fisher Chemical, United Kingdom); methanol 99% purity (Fisher Chemical, Loughborough, UK); formic acid 98% purity (Fluka, Seelze, Germany), trifluoroacetic acid (TFA) (Sigma-Aldrich, Saint Quentin Fallavier, France); monobasic potassium phosphate (Sigma-Aldrich, Saint Quentin Fallavier, France). All reagents were of certified HPLC grade.

### 2.2. Methods

#### 2.2.1. Study Design

Prior to the drug loss (sorption) studies, and in order to assess potential chemical composition differences between materials, the inner surfaces of all the tested catheters (before use) were characterized by Attenuated Total Reflectance Fourier Transform Infrared Spectroscopy (ATR-FTIR) and by surface Zeta potential measurements.

Sorption studies were first conducted to evaluate the impact of different individual catheters on drug loss during dynamic contact, simulating an 8 h infusion. Secondly, each drug was infused through a complete setup, including a syringe, an extension set, and a catheter simulating an 8 h infusion at two different flowrates (1 mL/h and 10 mL/h). The drug concentrations were representative of their respective clinical use conditions. The dilutions were performed as described in Section 2.1.2 to reach the following concentrations: paracetamol 1 mg/mL, diazepam 0.2 mg/mL and insulin 0.1 UI/mL. The pH was not adjusted after dilution and was found to be of 5.3 for paracetamol, 5.4 for diazepam, and 6.4 for insulin.

#### 2.2.2. Evaluation of Drug Loss Caused by Individual Catheters

Identical protocols to those previously published for the study of individual extension sets were used [12].

The intravenous (IV) infusion simulations (dynamic contact) were carried out using an electric syringe pump (Orchestra^®^ DPS modules, Fresenius, France), at two different flowrates: 1mL/h and 10 mL/h, which are flowrates commonly used for IV drug infusion.

The catheters were directly plugged to the syringe. The samples of the drug solution were collected from the tip of the syringe before contact with the tubing (Ti), then at T0 at the end of the catheter, after purging. Other samples were collected at the end of the catheter without stopping the infusion, after 1, 2, 4, and 8 h of simulated infusion (expressed as T1, T2, T4, and T8). An approximate volume of 150 µL was collected for each analysis time (minimum volume needed to perform the quantitative analysis) and, thus, the sampling time was flowrate dependent (about 1 min and 10 min respectively for the 10 mL/h and 1 mL/h condition). Visual control and Active Pharmaceutical Ingredient (API) quantification were performed on the samples.

#### 2.2.3. Sorption Studies of Complete Infusion Lines

Two setups of complete infusion lines were tested (Table 2) and were composed of a syringe, an extension set and a catheter. Among the extension sets previously studied, two of them (PVC and PE/PVC) had previously showed to behave quite differently [12]. Indeed, PVC extension sets had a strong tendency to cause sorption, whereas PE/PVC extension sets generally generated much less loss of API. A monolayered PVC extension set was used in setup 1 while a PVC coextruded with PE (inner part) was used in setup 2. The selection of the medical devices used in the complete infusion setup was based on the results of the sorption studies of individual MD and also took into account the clinical use of the medical devices (see Section 3.2.2, bullet point medical device selection).

An infusion of each drug was simulated in both dynamic conditions (1 and 10 mL/h) for both setups. The experimental setup simulating a complete infusion line is presented in Figure 1. Samples were collected only at the end of the infusion line. Sampling methodology followed the same protocol as for individual MD.

### 2.3. Analysis

#### Analysis of Catheters Inner Surface

Attenuated Total Reflectance Fourier Transform Infrared Spectroscopy (ATR-FTIR)

ATR-FTIR spectra of the inner surface of each catheter were acquired with a spectrum 100 spectrometer (PerkinElmer, Villebon sur Yvette, France) equipped with an ATR diamond crystal. All spectra were acquired from 3500 to 650 cm^−1^, using 16 scans with a 2 cm^−1^ resolution. 

Surface Zeta Potential

In contact with an aqueous solution, a solid surface assumes a surface charge. The Zeta potential (or electrokinetic potential) describes the charging behavior at interfaces. Surface Zeta potential is representative of the electric charge at the shear plane between the diffuse layer and the immobile layer of a material. The surface Zeta potential of the inner surface (before any drug administration) of all tested IV-tubings was assessed by measuring the streaming potential with a Surpass 3 (Anton Paar, Les Ulis, France) equipped with a tubing cell analysis system, in a 1 mmol/L potassium chloride solution at pH 5 before analysis in order to standardize the conditions. As the catheters were too small in diameter to be analyzed as such by the tubing cell analysis system, adequate shrinkable tubings provided by Anton Paar were used to insure a correct connection of the catheters in the cell.

Quantitative Analysis

The concentration of API in the infused solution was quantified by using a high-pressure liquid chromatography system (LC-2010HT compact system, Shimadzu, France). Analytical methods and validation data of each API are presented in previously published work [12]. All three methods possessed mean accuracy, repeatability and intermediate precision coefficients lesser than 5.0%, excepted for insulin for which a slightly more important variability was noted (mean intermediate precision coefficient of 6.4%).

All samples were diluted to within theoretical calibration curve range, and if beneath quantification limit the samples were reanalyzed after adapting the dilution.

Expression of the Results

For all three tested API, the results were expressed as the recovered percentage of the initial concentration (measured at Ti). Error bars expressed the 95% confidence interval of the mean value.

As all the tested catheters had different lengths and inner diameters, the results of API quantification was made comparable from one catheter to one another, by dividing the remaining percentage of the initial concentration by the surface contact area of the tubing. Sorption rate standardized by area of contact between drug solution and tubings inner material were calculated with Equation (1), and expressed as a percentage of sorption per square centimeter of tubing.
(1)$$\mathrm{Sorption}=\left(1-{\mathrm{Ratio}}_{Recovered\_catheter}\right)\times \frac{1}{S}\times 100$$

Ratio*_Recovered_catheter_*: ratio of API recovered at the end of the catheter at the analysis time over the initial concentration. *S*: inner surface area (cm^2^).

Since the loss due to both extension sets was previously evaluated [12], the dosage was only performed at the end of the infusion line. However, the loss due specifically to the catheter (final MD of the infusion line) was estimated using Equation (2), so that it could be compared to the loss induced by the catheter alone.
(2)$${\%}_{loss\_catheter}=100-{\%}_{loss\_ext}-{\%}_{recovered}$$

%*_loss_catheter_* = percentage of API loss due to the catheter only. %*_loss_ext_* = percentage of API loss due to the extension set only (data obtained from [12]). %*_recovered_* = percentage of API recovered from initial concentration at the end of the setup.

Statistical Analysis

All statistical analyses were performed using Stata statistical software (version 15, StataCorp, College Station, TX, USA). Continuous parameters were expressed as mean and standard-error of mean (SEM) according to statistical distribution. The assumption of normality was studied using Shapiro-Wilk’s test.

Hedge’s effect size was calculated as described in (Equation (3)). All the effect sizes were calculated from the standardized values calculated with Equation (1).

Statistical analysis was performed by calculating the Hedge’s effect size (Equation (3)).
(3)$$ES=\frac{m_{1}-m_{2}}{SD_{pooled}}=\frac{m_{1}-m_{2}}{\sqrt{\frac{\left(n_{1}-1\right)s_{1}^{2}+\left(n_{2}-1\right)s_{2}^{2}}{n_{1}+n_{2}-2}}}$$

$m_{1}\;\mathrm{and}\;m_{2}$: mean at T8 for Turbo-Flo^®^ catheters ($m_{1}$) and other catheters ($m_{2}$. $n_{1\;}\mathrm{and}\;n_{2}$: sample sizes, and $s_{1}\;\mathrm{and}\;s_{2}$: standard deviations.

For a given catheter, a positive effect size was interpreted as a less important tendency to sorption compared to the Turbo-Flo^®^ reference, while a negative size effect was interpreted oppositely as a more important tendency to promote sorption. If 0 was included in the confidence interval, the result was interpreted as non-significant. Forest-plots were used to represent graphically these results.

## 3. Results

### 3.1. Catheter’s Surface Characterization

#### 3.1.1. ATR-FTIR

The analysis of the internal surface of the catheters by infrared spectroscopy (Figure 2) highlighted a difference of composition between the catheters. The spectra obtained for the Blue FlexTip^®^ and PowerPicc^®^ catheters showed a high degree of similarity in favor of a very similar composition. The Turbo-Flo^®^ catheters exhibited a different spectrum from the other two PU catheters, particularly showing three new bands at 1739, 1717, and 1244 cm^−1^. The spectra of the Lifecath^®^ catheters was consistent with a silicone spectrum.

#### 3.1.2. Surface Zeta Potential

The surface charge at the internal surface of the catheters was assessed by measuring surface Zeta potential. Table 3 presented Zeta potentials at a pH close to 5.0. When comparing PUR catheters (Blue FlexTip^®^, PowerPicc^®^, and Turbo-Flo^®^), Blue FlexTip^®^ and PowerPicc^®^ exhibited Zeta potential values close to each other, but for Turbo-Flo catheters, a lower zeta potential was observed.

### 3.2. Drug loss Studies

#### 3.2.1. Individual Catheters

1 mL/ Dynamic Condition

During the simulation of a 1 mL/h infusion, no significant loss of paracetamol was highlighted with any of the studied catheters (Figure 3A). However, variable losses of diazepam were observed (Figure 3B). At T0, diazepam concentrations were significantly reduced (loss >60%) for all the tubings except the PowerPicc^®^ catheters, which still induced important losses of diazepam but fared a little better than the three other catheters. The most important loss was observed for silicone tubing (Lifecath^®^). The loss profile was similar for all tubings with the lowest concentration reached at T1. However, taking into account the internal contact surface (Appendix A), Blue FlexTip^®^ and Lifecath^®^ catheters exhibited sorption of 8.9 ± 0.2 %/cm^2^ and 8.4 ± 0.1 %/cm^2^ at T8, respectively, while the PowerPicc^®^ and Turbo-Flo^®^ catheters exhibited lower sorption ratios (4.9 ± 0.1 %/cm^2^ and 6.0 ± 0.1 %/cm^2^, respectively).

During insulin infusion (Figure 3C), a loss of API of approximately 15% was observed at T0 for all catheters, but several kinetic profiles could be distinguished. The Blue FlexTip^®^ and PowerPicc^®^ catheters showed a maximum loss at T1, then returned to a concentration close to initial concentration at T2 but decreased again up to T8 (39.7 ± 9.2% and 31.9 ± 7.9% respectively). The Turbo-Flo^®^ catheters also showed a maximum loss at T1, then the concentration increased again before stabilizing at a value closer to that at T0 (28.8 ± 3.0% loss at T8). Finally, the silicone catheters (Lifecath^®^) had a different profile, as insulin concentrations decreased steadily with time until T8 (37.0 ± 8.3% loss).

10 mL/h Dynamic Condition

During the 10 mL/h dynamic contact experiments, paracetamol concentrations remained stable (Figure 4). The concentration of diazepam decreased in contact with each catheter, but the loss due to sorption was less important when compared to the 1 mL/h infusion. Similarly, the loss of insulin by sorption was lower with the 10 mL/h infusion than with the 1 mL/h infusion.

Effect Size (ES)

The effect size at T8 was calculated using Turbo-Flo^®^ catheters as the reference tubing (Figure 5), as it was the one chosen in the complete setup (as explained later in Section 3.2.2, bullet point medical device selection). No significant variation (ES confidence interval including 0) between all the tested catheters was observed after paracetamol and insulin infusion at 1 mL/h. As effect size was positive after paracetamol infusion at 10 mL/h, sorption was the most important with the Turbo-Flo^®^ catheter in this condition. For diazepam infusions, the ES was positive only for PowerPicc^®^ catheters (comprised between 6.1 and 29.9 at 1 mL/h and between 1.7 and 9.4 at 10 mL/h). Overall, diazepam loss by sorption was significantly more important with Blue FlexTip^®^ and Lifecath^®^ than with Turbo-Flo^®^ catheters, but less for PowerPicc catheters.

#### 3.2.2. Complete Infusion Setup

Medical Device Selection

Regarding sorption ratios with all catheters, the two PICC behaving similarly (Turbo-Flo^®^ was slightly more prone to sorption than PowerPicc^®^) behaved better than the Blue FlexTip^®^ and Lifecath^®^. Based on the annual estimated consumption of catheters in our hospital, the use of single lumen PICC-lines is twice higher than single lumen CVC and about 40 times higher than tunneled silicone CVC. Among the catheters, we chose a PICC-line because of the frequent use of this type of catheter in clinical practice. After analysis of the sorption induced by the different catheters, the Turbo-Flo^®^ PICC was selected for use in the complete setup.

The syringes that were selected for the study were classical syringes used with electric syringe pumps in our hospital. Sorption studies were also performed on individual syringes and the results are presented in Appendix A.

1 mL/h Dynamic Condition

In the simulated low-flow infusion (1 mL/h), no significant variations from the initial paracetamol concentration were observed (Figure 6A). During diazepam infusion (Figure 6B), extreme losses of API were observed for both sets. In Set 1 (incorporating a PVC extension set), API concentrations decreased continuously to a maximum loss at T8 of 96.4 ± 0.9% from the initial concentration. In the case of setup 2 (incorporating a PE/PVC extension), the loss remained roughly constant from T1 to T8 and was between 83.4% and 90.9%.

During insulin infusion, API concentrations also decreased during infusion with the two tested setups (Figure 6C). The loss of API over time followed a similar kinetic pattern: a minimum was reached at T1 then concentrations raised from T2 onwards and reached a plateau, but at a much lower concentration than the initial concentration. A greater loss was observed with setup 2 (66.1 ± 1.2%) when compared with setup 1 (54.0 ± 5.6%).

10 mL/h Dynamic Condition

As shown in Figure 7A, no variations in paracetamol concentration were observed during the infusion via the two setups at the 10 mL/h flow rate.

Losses of diazepam and insulin were observed during the 10 mL/h infusion, but they were overall of lesser intensity than for the 1 mL/h flowrate. For diazepam (Figure 7B) with setup 1, a maximum loss was observed at T1 (84.0% loss) and then the concentrations increased slightly up until T8 (72.6% loss). A similar kinetic pattern was observed for setup 2, however the loss of API was greatly reduced (losses ranging from 30.8% at T1 to 17.1% at T8). During insulin infusion, for both setups, a maximum loss was reached at T1 (27.4 ± 2.7% for setup 1 and 56.0 ± 2.0% for setup 2) and then the concentrations raised back to a value close to the initial concentration and remained stable.

Setup Comparison

The recovered percentage of API with isolated medical devices is shown in regard with the recovered percentage of API infused through the complete infusion setup in Figure 8. The comparisons showed that the final loss was mainly due to the medical device with the highest loss (superimposition of the orange curve with the lowest diagram). In addition, the diazepam study (Figure 8B) showed that at a flow rate of 1 mL/h, setups 1 and 2 induced similar losses, whereas the individual extension tubings had very different losses of active ingredient. This result highlighted that, in the case of setup 1, the loss was mainly due to the extension tubing and the loss induced by catheter did not had a strong impact on the overall sorption. On the other hand, in setup 2, the loss due to the extension set alone is much lower, so the final loss is mainly due to the catheter. Similar phenomena were also observed with diazepam at the 10 mL/h rate and with insulin at both rates.

As summarized in Figure 9, these findings indicate that the catheter induced different losses when tested alone and when included in an infusion line. For medicines that exhibited loss of active ingredient by sorption (diazepam and insulin), the loss due to the catheter in the complete infusion line was less than the loss it caused when alone. This result revealed a low loss due to the catheter when it was preceded by an extension set with a strong tendency to sorption. On the other hand, when the extension set caused a smaller decrease in concentration, the catheter loss was similar to that observed with isolated catheters.

## 4. Discussion

In this work, diazepam, insulin, and paracetamol were used as models to the study the drug loss through sorption that could be caused by catheters. The sorption behavior of these models has already been well characterized. Diazepam acted as a model drug for absorption [8,11,16,19,20], insulin as a model of adsorption [21,22,23], and paracetamol as a low sorption model drug. The first part of this study highlighted that a loss of API due to sorption occurred with all the tested catheters. In the second part, two complete infusion setups were compared and focused on the impact on the overall sorption when changing the extension set, comparing a highly sorptive tubing and a lower one. Both parts of this work highlighted that catheters are critical medical devices when considering sorption issues.

When studied individually, all the catheters we tested induced significant losses of diazepam and insulin, which is coherent with previous studies showing the tendency of PUR tubings in medical devices to induce sorption (6,7,12,13,15). Based on FTIR spectroscopy analysis, two of the catheters were made of similar polyurethanes (Blue FlexTip^®^ and PowerPicc^®^), but of different internal surface areas (due to different length and inner diameter). Unexpectedly, the catheter with the lowest surface area (Blue Flex Tip^®^ central venous catheter) was found to cause the higher loss. This is surprising as it has already been shown that the longer the tubing in contact with the solution the more sorption can be generated [13]. Even though no differences were observed in their general composition, other factors could impact their capacity to induce sorption, like a difference of their extreme surface (the first angstroms) which could not have been detected by ATR-FTIR: difference in surface roughness or the presence of additional compounds in reason of their fabrication process (slip agents for example) or storage conditions (migration of additives or polymer modification). Two PICCs consisting of two PUR’s with different chemical compositions (based on their infrared spectra) were also studied and presented different sorption profile. The difference in composition of the PowerPicc^®^ and TurboFlo^®^ can be an explanation for the difference between these tubings of measured surface Zeta potential. The influence of Zeta potential on sorption capacity was previously hypothesized [12], and this result was congruent with this previous finding as the closest to zero Zeta potential was responsible for the most important loss among PUR catheters. Silicone rubber is also known to present a risk of sorption [11,18]. Among all of the tested catheters, the silicone one appeared to have the highest tendency to induce diazepam sorption and was also correlated with a low surface Zeta potential. In a previous study, silicone catheters exhibited a higher water contact angle than PUR catheters [24], indicative of a higher lipophilicity. Thus, the affinity of lipophilic drugs, such as diazepam for silicone catheters, might be higher than for PUR catheters.

To investigate the impact of a whole infusion line on sorption phenomena, two setups using a syringe, an extension set, and a catheter were studied. The two selected setups differed in the extension used, the syringe and catheter being common to both setups. The Turbo-Flo^®^ catheter was used as the reference catheter for effect-size calculations as it was the most used single lumen PICC-line in our hospital and because based on individual MD sorption results it presented a better profile than the Blue FlexTip^®^ catheter and a similar one to that of the PowerPicc^®^ catheter (see Section 3.2.2). Thus, the comparison of these two setup allowed us to evaluate the effect of replacing an extension set with a high sorption potential (setup 1 with a PVC extension set) by an extension set with a lower potential (setup 2 with a PE/PVC extension set). In order to optimize the clinical setup, it has been suggested to decrease the length of tubing used [25] and consider the material used even if the length is short [5]. The results of this study showed that changing the extension set did not optimize the whole infusion line and showed that the impact of the catheter on sorption phenomena depended on the devices preceding it in the infusion line. Indeed, when high API loss by sorption occurred before the catheter, the catheter induced little additional API loss. On the opposite, when the API loss before the catheter was low, the catheter had an important role and was mainly responsible for the loss. In order to optimize an infusion set in terms of risk of sorption, it is not sufficient to modify the extension line, but it will also be necessary to improve the catheter. The data presented in this work showed that, for the duration of the study, the replacement of the extension set had little impact on the overall sorption, even though the extension set was the medical device with the highest surface contact area. The loss caused by the complete setup followed the same trend as the loss induced by most absorptive tubing in the setup but was not equal to the cumulated loss induced individually by the extension sets and the catheters. It is therefore impossible to precisely deduce complete drug sorption risks possibly caused by a complete setup only by studying individual devices. Indeed, as highlighted by our results, the total drug sorption calculated by summing-up the sorption caused by individual medical devices would be higher than the real impact in a whole clinical setup. It is also important to consider the position of a medical device in the setup to evaluate its risk potential.

Adsorption is a very fast phenomenon, but absorption take longer to reach equilibrium, as shown for example in the study published by Al Salloum et al., in which they showed that the absorption of diazepam by plasticized PVC could take more than 70 h to reach equilibrium during static contact [16]. This is coherent with the results observed in our experimental setup simulating an 8-h infusion (which is representative of a long infusion in clinical conditions), as the equilibrium was clearly not reached for diazepam at neither flow rates (the remaining diazepam concentrations remained lower than 25% of the initial concentration (Figure 6B and Figure 7B). Being a much larger molecule, insulin was expected to only adsorb (and not absorb) to the surface, and therefore could potentially quickly saturate the surface. Indeed, this was clearly seen when it was infused at 10 mL/h, as insulin concentrations regained their initial levels after only 2 h of infusion (Figure 7C), but this was not observed for a flow rate of 1 mL/hour (Figure 6C), thus indicating that the flow of insulin didn’t bring enough insulin to saturate the surface, even after 8 h. It can also be noted that in the insulin formulation, excipients (such as phenol or m-cresol) are also prone to absorption (for example to plasticized PVC, as shown in a study published by Masse et al. [4]) and this could also influence the equilibrium by competition between the molecules. The exact property or combination of properties that is responsible for sorption interactions are not completely known. The lipophilicity of a molecule has often been cited as a main factor for API/material interactions [26,27,28]. Indeed, Ziccardi et al. reported that the polymer–water partitioning coefficient of numerous organic compounds was correlated with their octanol–water partition coefficients (LogP) [29]. However, other factors also seem to play a part, like the net charge of the molecule, which is linked to functional groups and the pH of the solution. For example, Illum et al. showed that the loss of warfarin (a weak acid with a pKa of 5.01) varied when the pH ranged from 2 to 7.5 [30]. As such, it is not possible to link the sorption capacity of a compound specifically to any functional group.

Drugs with high lipophilicity are therefore most likely to be impacted by sorption phenomena [30,31]. The clinical impact of drug sorption can be important with anticancer or narrow therapeutic index drugs, such as immunosuppressive drugs, e.g., tacrolimus (which has a known tendency for sorption [9,32]). Comparatively, studies evaluating the sorption capacity of monoclonal antibodies with infusion medical devices are scares, but despite that class of drug being known to be at risk of adsorption with different materials [33,34,35,36], a recent study of our research team showed that bevacizumab, a monoclonal antibody used in oncology, did not seem to suffer any detectable concentration loss by sorption when infused through a complete infusion line including an infusion set and an implantable port equipped with either a silicone or a PUR catheter [24]. We explained this finding by the high concentrations used clinically (6 mg/mL), which possibly masked low level of sorption. However, infusions monoclonal antibodies at lower concentrations could possibly be at risk and would need to be more thoroughly investigated. As an example of a possible option, administration bags of blinatumomab (Blincyto^®^) must be prepared using a protective coating solution of polysorbate 80 to prevent protein adsorption [37].

In this work, the complete infusion setup tested was the simplest setup possible, with only a syringe, an extension set and a catheter. In clinical practice, and not only in critical care units, infusion setups can be much more complex, containing many different medical tubings or other devices like filters that are known to cause sorption of lipophilic compounds [38,39,40,41]. The risks of sorption may therefore be even worse than what was assessed in this study, and further investigations are needed to fully evaluate the impact of complex infusion lines on drug sorption and patient care.

The effect of the pH was not investigated in this study, as we wanted to evaluate drug loss in a realistic clinical setting. However, this is a very important point, as the net charge of the APIs depends on it. At the pH of the medication solutions (paracetamol: pH = 5.3; diazepam: pH = 5.4; insulin: pH = 6.4), paracetamol and diazepam would be in unionized form but insulin would be positively charged. This could explain (at least partially) why insulin interacts with the negatively charged surface of the catheters (by a weak charge interaction). However, the combination of charge and important steric hindrance would not be in favor of its diffusion inside the polymer material, leading more plausibly exclusively to surface adsorption. If the pH of the medication or infusion solution were to be alkalinized, it is also possible that the interaction profile of insulin would be different. To date, the drug/material interface remains quite challenging to characterize, even if the use of molecular simulation is opening up new fields of research to help understand the interactions [42], the models generated still need to be validated by experimental analyses, including surface studies. In this study, the use of Zeta potential measurements was an innovative approach in the field of drug content-container interactions. Its preliminary use here could in the future be completed by dynamic adsorption tests (by following the surface Zeta potential whilst gradually increasing API quantity in contact with the tested material) as the adsorption of molecules onto the surface could modify the surface charge and be highlighted by a change of the surface Zeta potential.

## 5. Conclusions

This work used model drugs, such as paracetamol, diazepam, and insulin to investigate drug losses during infusions. It highlighted that silicone and PUR catheters can be at the origin of extremely high losses of diazepam and insulin, caused by sorption phenomena. This could potentially expose the patients to incomplete drug administration and under-dosing. Replacing a PVC extension set by a PE/PVC extension set, less prone to sorption, had no significant impact on the final drug loss in the duration of the test because of the sorption that was then caused by the catheter. The quality of the tubings used outside the patient (infusion and extension sets) has been greatly improved during the past few years and some of them are now manufactured to limit sorption risks, but a similar work still needs to be done on catheters in order to minimize drug losses during infusions.

## Figures and Tables

**Figure 1 pharmaceutics-13-01709-f001:**
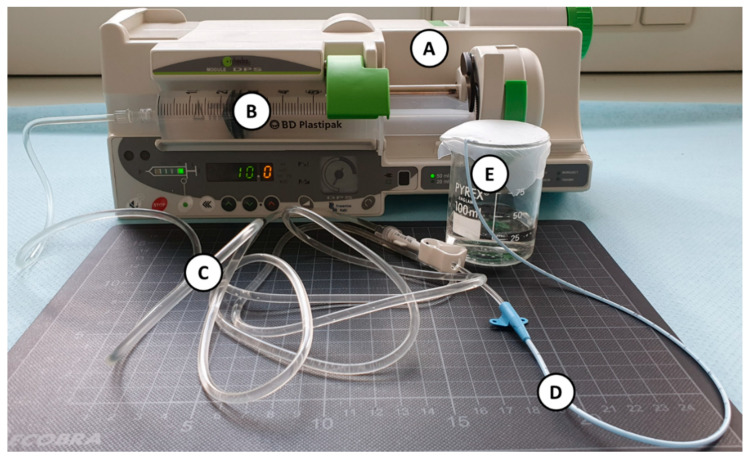
Picture of the experimental setup in dynamic condition (**A**): electric syringe pump; (**B**): 50 mL syringe; (**C**): extension set tubing; (**D**): catheter; (**E**): sampling site at the end of the infusion line).

**Figure 2 pharmaceutics-13-01709-f002:**
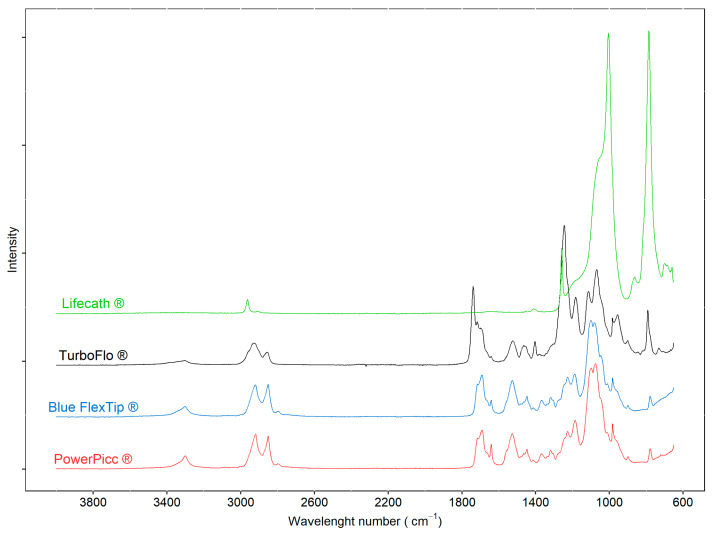
ATR-FTIR spectra of the inner surface of polyurethane (Blue: Blue FlexTip^®^; Red: PowerPicc^®^; Black: Turbo-Flo^®^) and silicone catheters (Green: Lifecath^®^).

**Figure 3 pharmaceutics-13-01709-f003:**
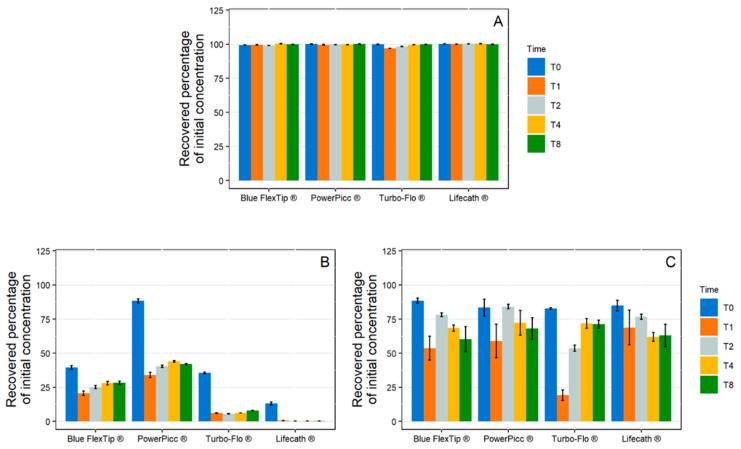
Evolution of the recovered percentage of initial concentration in paracetamol (**A**), diazepam (**B**), and insulin (**C**) in 1 mL/h dynamic condition with polyurethane catheters (Blue FlexTip^®^, PowerPicc^®^, and Turbo-Flo^®^) and silicone catheters (Lifecath^®^). (*n* = 3, mean ± standard error of the mean). T0–T8: different analysis times: immediately after purging (T0), then after 1 h (T1), 2 h (T2), 4 h (T4), and 8 h (T8) of infusion.

**Figure 4 pharmaceutics-13-01709-f004:**
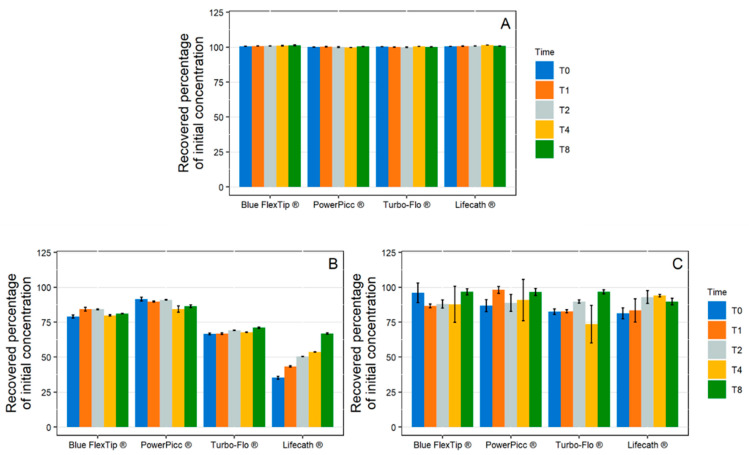
Evolution of the recovered percentage of initial concentration in paracetamol (**A**), diazepam (**B**), and insulin (**C**) in 10 mL/h dynamic condition with polyurethane catheters (Blue FlexTip^®^, PowerPicc^®^, and Turbo-Flo^®^) and a silicone catheter (Lifecath^®^). (*n* = 3, mean ± standard error of the mean). T0–T8: different analysis times: immediately after purging (T0), then after 1 h (T1), 2 h (T2), 4 h (T4), and 8 h (T8) of infusion.

**Figure 5 pharmaceutics-13-01709-f005:**
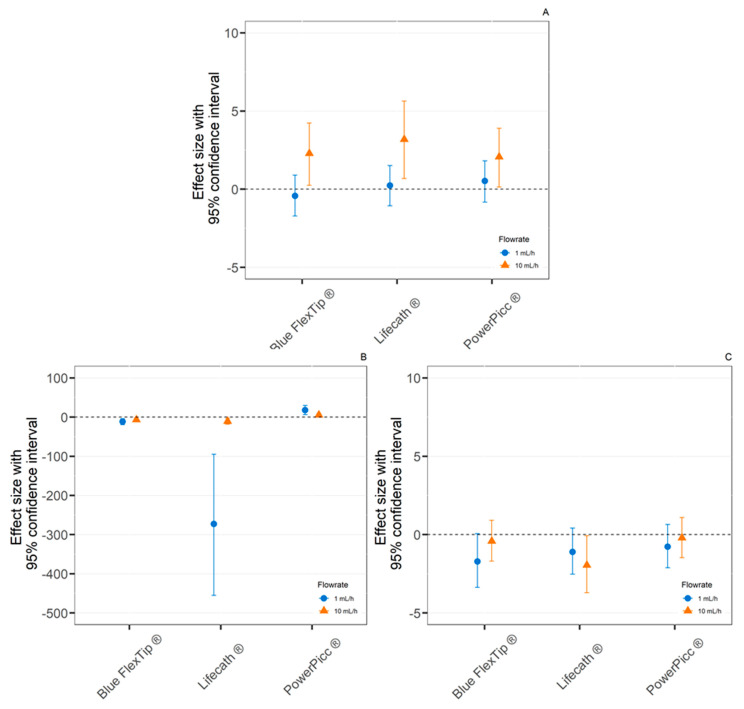
Effect size of the comparison between Turbo-Flo^®^ catheter and other studied catheters after an 8h infusion at 1 mL/h and 10 mL/h of paracetamol (**A**), diazepam (**B**), and insulin (**C**). (*n* = 3, mean ± 95% confidence interval).

**Figure 6 pharmaceutics-13-01709-f006:**
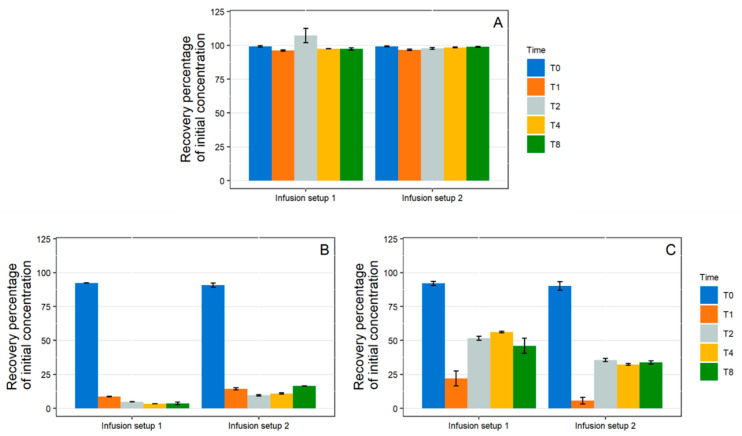
Evolution of the recovered percentage of initial concentration in paracetamol (**A**), diazepam (**B**), and insulin (**C**) in 1 mL/h dynamic condition with both complete infusion setup (setup 1: polypropylene syringe + polyvinyl chloride extension set + Turbo-Flo polyurethane catheter. Setup 2: polypropylene syringe + polyethylene coextruded with polyvinyl chloride extension set + Turbo-Flo polyurethane catheter (*n* = 3, mean ± standard error of the mean). T0–T8: different analysis times: immediately after purging (T0), then after 1 h (T1), 2 h (T2), 4 h (T4), and 8 h (T8) of infusion.

**Figure 7 pharmaceutics-13-01709-f007:**
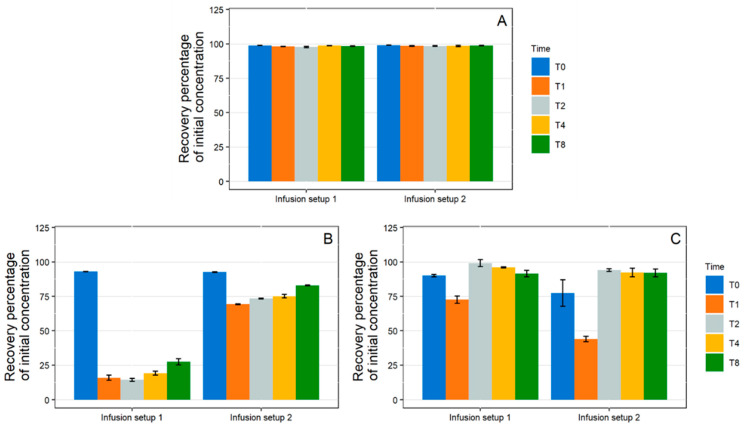
Evolution of the recovered percentage of initial concentration in paracetamol (**A**), diazepam (**B**), and insulin (**C**) in 10 mL/h dynamic condition with both complete infusion setup (setup 1: polypropylene syringe + polyvinyl chloride extension set + Turbo-Flo polyurethane catheter. Setup 2: polypropylene syringe + polyethylene coextruded with polyvinyl chloride extension set + Turbo-Flo polyurethane catheter (*n* = 3, mean ± standard error of the mean). T0–T8: different analysis times: immediately after purging (T0), then after 1 h (T1), 2 h (T2), 4 h (T4), and 8 h (T8) of infusion.

**Figure 8 pharmaceutics-13-01709-f008:**
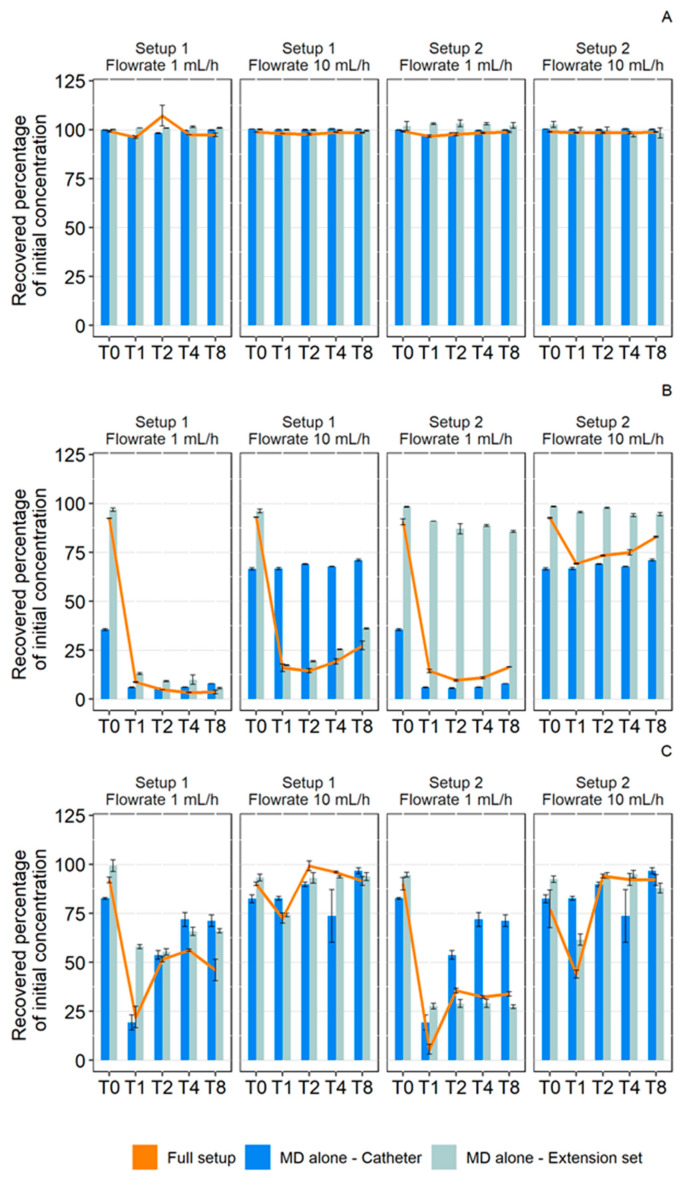
Comparison of the recovered percentage of initial concentration in paracetamol (**A**), diazepam (**B**), and insulin (**C**) during 1 mL/h and 10 mL/h dynamic contact with medical devices (MD) alone (blue: catheter; gray: extension set) and with a complete setup (orange). (Setup 1: polypropylene syringe + polyvinyl chloride extension set + Turbo-Flo polyurethane catheter. Setup 2: polypropylene syringe + polyethylene coextruded with polyvinyl chloride extension set + Turbo-Flo polyurethane catheter (*n* = 3, mean ± standard error of the mean). T0–T8: different analysis times: immediately after purging (T0), then after 1 h (T1), 2 h (T2), 4 h (T4), and 8 h (T8) of infusion.

**Figure 9 pharmaceutics-13-01709-f009:**
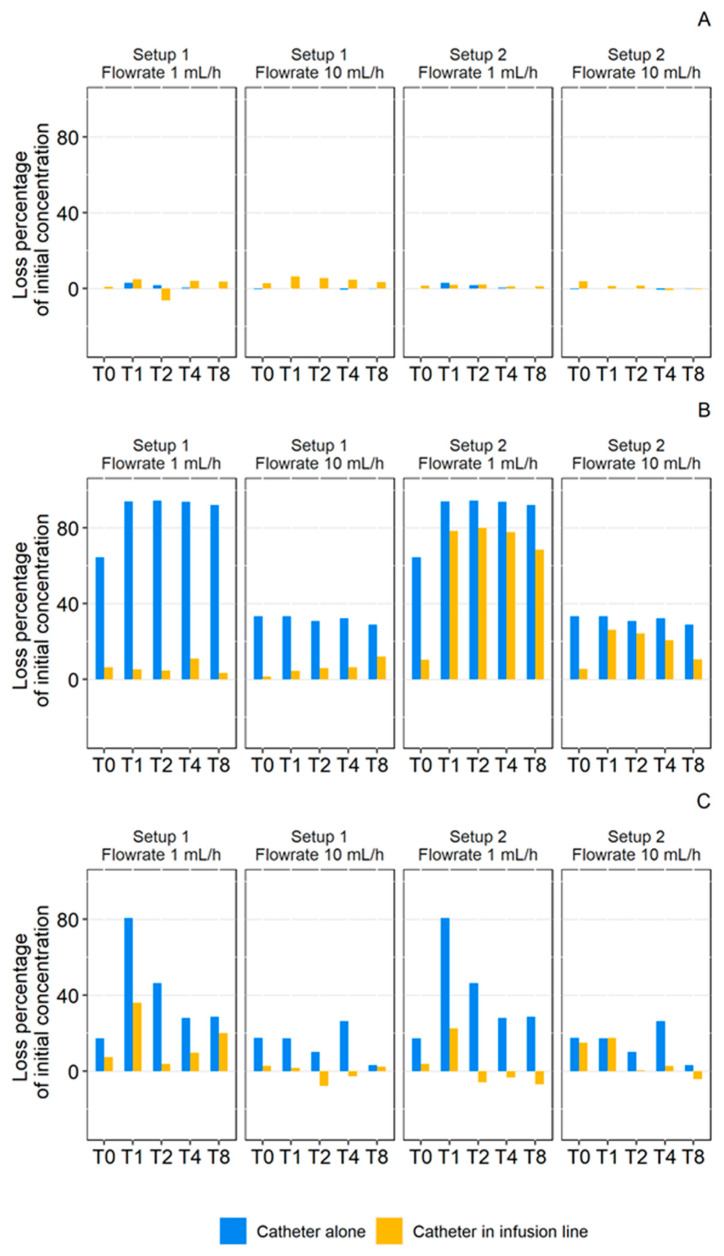
Comparison of the calculated loss percentage of initial concentration in paracetamol (**A**), diazepam (**B**), and insulin (**C**) during 1 mL/h and 10 mL/h through a catheter alone (Blue) or a catheter in an infusion line (Orange). (Setup 1: polypropylene syringe + polyvinyl chloride extension set + Turbo-Flo polyurethane catheter. Setup 2: polypropylene syringe + polyethylene coextruded with polyvinyl chloride extension set + Turbo-Flo polyurethane catheter. T0–T8: different analysis times: immediately after purging (T0), then after 1 h (T1), 2 h (T2), 4 h (T4), and 8 h (T8) of infusion.

**Table 1 pharmaceutics-13-01709-t001:** Summary of the studied medical devices and their characteristics.

	Product Code	Material	Length (cm)	Inner Diameter (cm)
**Syringe**				
Plastipak^®^, Becton-Dickinson (Le Pont de Claix, France)	300865	Barrel: polypropylene Plunger rod: polypropylene Seal: synthetic isoprene	Barrel: 13.3	Barrel: 2.65
**Catheters**				
Blue FlexTip^®^ central venous catheter, Teleflex Medical (Le Faget, France)	CV-04301	Polyurethane	20.0	0.13
Power Picc^®^, BARD Medical (Voisins Le bretonneux, France)	6175118	Polyurethane	40.0	0.094
Turbo-Flo^®^, COOK Medical^®^ (Paris, France)	G12987	Polyurethane	40.0	0.12
Lifecath^®^, VYGON (Ecouen, France)	2191.50	Silicone	40.0	0.095
**Extension sets**				
CAIR LGL (Lissieu, France)	PN10318-1	Polyvinyl chloride	2000.0	0.25
CAIR LGL (Lissieu, France)	RPB5320	Outer layer: Polyvinyl chloride Inner Layer: Polyethylene	2000.0	0.25

**Table 2 pharmaceutics-13-01709-t002:** Composition of the two studied complete infusion setups (PP: polypropylene; PVC; polyvinyl chloride; PE; polyethylene; PUR: polyurethane).

	Infusion Setup 1		Infusion Setup 2	
Medical Device	Manufacturer	Material	Manufacturer	Material
Syringe	Plastipak^®^ (Becton-Dickinson)	PP	Plastipak^®^ (Becton-Dickinson)^®^	PP
Extension set	CAIR LGL	PVC	CAIR LGL	PE/PVC
Catheter	Turbo-Flo^®^ (Cook Medical)	PUR	Turbo-Flo^®^ (Cook Medical)	PUR

**Table 3 pharmaceutics-13-01709-t003:** Zeta potential of polyurethane (Blue FlexTip^®^, PowerPicc^®^, and Turbo-Flo^®^) and silicone catheters (Lifecath^®^).

	Blue FlexTip ^®^	PowerPICC ^®^	Turbo-Flo ^®^	Lifecath ^®^
pH	5.0	5.0	4.9	5.1
Zeta potential (mV)	−30.0	−25.2	−11.8	−32.6

## Data Availability

Full raw data are provided in the Appendix A.

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
