# Peer review of "Critical Drug Loss Induced by Silicone and Polyurethane Implantable Catheters in a Simulated Infusion Setup with Three Model Drugs"

_pharmaceutics, 2021, doi:10.3390/pharmaceutics13101709_

Round 1
Reviewer 1 Report
The manuscript is interesting Critical drug loss induced by silicone and polyurethane implantable catheters in a simulated infusion setup with three model drugs is an interesting manuscript but required major revision;
- Lack of references in the entire manuscript. Provide the reference at appropriate places such equations 1, ,2, 3, introductions. Etc……..
- Insulin is a peptide; did you have any recommendation between small molecule (Diazepam, paracetamol) Vs large molecule.
- What drug property is responsible for sorption/drug loss. Any functional group?
- Did effect of pH also studied?
- On page 2 line 65, 66, 67 - This study aimed therefore to assess the drug loss by sorption that could be induced individually by four commonly used catheters, and evaluate the impact of a complete in fusion line, composed of a syringe, an extension set and a catheter on the amount of drug that would be retained by the MD during simulated infusions. What do you mean by retained by MD.
- On page 4, line 104, 105: Prior to the drug loss (sorption) studies, the inner surface of all the tested catheters (before use) was characterized for…………………… by Attenuated Total Reflectance Fourier Transform Infra.
- On page 4 line 132- 134- Provide reference. The selection of the medical devices used in the complete infusion setup was based on the results of the sorption studies of individual MD and also took into account the clinical use 134 of the medical devices.
- Line 170, Anton Paar: Provide details of city/company.
- It is helpful for readers to put T1, T2, T3…………. In tabular form and comments
- Line 296: (as explained later in para-296 graph Chapitre 1 :1.1.1.a)). I didn’t find that?
- Line 410, Any explanation for this; Unexpectedly, the catheter with the lowest surface area (Blue Flex Tip® central 410 venous catheter) was found to cause the higher loss.
- explain any finding or relation with sorption and zeta potential in discussion section
- page 17: This study highlighted that silicone and PUR catheters can sometimes be at the origin 477 of extremely high drug losses caused by sorption phenomena, therefore potentially ex-478 posing the patients to incomplete drug administration and under-dosing. What do you mean by sometimes? At what occasions? Rewrite the conclusion and explain in detail since this is a relative study with three different drugs.
- I would recommend to summarize your finding in the tabular form.
- Explain the figure legends for all the figures. It is extremely helpful for the readers.
Reviewer 2 Report
The work shows how the material of the catheter affects the final concentration of the substance passing through them. The relevance of the work is beyond doubt, since it is practically important. The results are well processed and statistical calculations are presented.
Reviewer 3 Report
This manuscript reports on drug loss induced by silicone and polyurethane catheters in an infusion setup. Three model drugs are used for evaluation. It is interesting, but this paper is mainly described phenomenologically. Reviewer thinks more data-based discussion is need. The following points should be considered.
- Sorption to devices depends on the concentration of the test solution. The concentrations of three model drugs in the sorption study and evaluation of drug loss by catheters should be written.
- In the drug loss study, why do the recovered percentage change with time? Doesn’t the saturation of sorption reach at T1? Is there any specific equivalent kinetics between drugs and devices? Please discuss carefully.
- What does the reference mean in Table 1? Product code or lot number?
- Reviewer thinks %recovered catheter is not correct on the equation 1. Ratiorecovered catheter seems to be correct. Please check this point.
- There is a typo in the equation 2.
- In the effect size discussion, why is Turbo-Flo® used as a reference?
- Inline 410 - 411, the authors describe “unexpectedly, the catheter with the lowest surface area (Blue Flex Tip® central venous catheter) was found to cause the higher loss.” Please discuss the reason of this phenomenon.
- Please discuss the result of the sorption of insulin in detail.
Round 2
Reviewer 1 Report
The authors replied to the comments satisfactorily. I appreciate the author's honest response for tabular changes and it is fine if that doesn't fit to the manuscript. The revised manuscript looks good, however, I have some minor corrections and questions for the authors,
1. Minor change:
Conclusion, line 538 lost some word and need minor edits to make this sentence meaningful……..
This study, using three model drugs, highlighted that silicone and PUR catheters can sometimes for some substances like diazepam and insulin be at the origin of extremely high drug losses caused by sorption phenomena, therefore potentially exposing the patients to incomplete drug administration and under-dosing.
2. In line 295, Equation 1, Why do authors change the formula from recovered to the ratio?
3. Still I do understand the meaning of MD? If you could provide a full name to an acronym.
Reviewer 3 Report
Reviewer still thinks Equation 1 is not correct. RatioRecoverd_catheter seems to be correct, not %Ratio_ctheter.
If percentage is used, 1-%Ratio_ctheter should be less than 1. So the value of Sorption will be negative. Please check this point.
